# Synapse-Mimetic Hardware-Implemented Resistive Random-Access Memory for Artificial Neural Network

**DOI:** 10.3390/s23063118

**Published:** 2023-03-14

**Authors:** Hyunho Seok, Shihoon Son, Sagar Bhaurao Jathar, Jaewon Lee, Taesung Kim

**Affiliations:** 1SKKU Advanced Institute of Nanotechnology (SAINT), Sungkyunkwan University, Suwon 16419, Republic of Korea; 2Department of Nano Science and Technology, Sungkyunkwan University, Suwon 16419, Republic of Korea; 3School of Mechanical Engineering, Sungkyunkwan University, Suwon 16419, Republic of Korea

**Keywords:** memristor, artificial intelligence, neuromorphic computing, synapse, convolutional neural network, resistive random-access memory, bioinspired device

## Abstract

Memristors mimic synaptic functions in advanced electronics and image sensors, thereby enabling brain-inspired neuromorphic computing to overcome the limitations of the von Neumann architecture. As computing operations based on von Neumann hardware rely on continuous memory transport between processing units and memory, fundamental limitations arise in terms of power consumption and integration density. In biological synapses, chemical stimulation induces information transfer from the pre- to the post-neuron. The memristor operates as resistive random-access memory (RRAM) and is incorporated into the hardware for neuromorphic computing. Hardware composed of synaptic memristor arrays is expected to lead to further breakthroughs owing to their biomimetic in-memory processing capabilities, low power consumption, and amenability to integration; these aspects satisfy the upcoming demands of artificial intelligence for higher computational loads. Among the tremendous efforts toward achieving human-brain-like electronics, layered 2D materials have demonstrated significant potential owing to their outstanding electronic and physical properties, facile integration with other materials, and low-power computing. This review discusses the memristive characteristics of various 2D materials (heterostructures, defect-engineered materials, and alloy materials) used in neuromorphic computing for image segregation or pattern recognition. Neuromorphic computing, the most powerful artificial networks for complicated image processing and recognition, represent a breakthrough in artificial intelligence owing to their enhanced performance and lower power consumption compared with von Neumann architectures. A hardware-implemented CNN with weight control based on synaptic memristor arrays is expected to be a promising candidate for future electronics in society, offering a solution based on non-von Neumann hardware. This emerging paradigm changes the computing algorithm using entirely hardware-connected edge computing and deep neural networks.

## 1. Introduction

According to Moore’s law, the performance of semiconductors will double every 24 months, and this has been maintained through the development of state-of-the-art foundry and chipmaker technologies [1,2,3,4,5]. However, the development of few-atom-scale semiconductor processes to achieve low-power operation with fast information processing has several limitations, including those imposed by Moore’s law [6,7,8,9,10,11]. Limitations of conventional computing technology include memory bottlenecks and high-cost energy processing (data processing between memory and processor). Moreover, the emerging artificial intelligence (AI) techniques require parallel information processing, big data analysis, and integrated systems entailing in-memory and on-chip computing [12,13,14,15,16]. However, conventional von Neumann architecture suffers memory bottlenecks as a result of continual data processing between the memory and processor, resulting in low-efficiency energy and low-speed memory processing [17,18,19]. Neuromorphic computing has been developed to overcome the memory bottleneck associated with von Neumann architecture [20,21,22,23,24]. Biological synapse-mimetic devices exhibit human-brain-like operations and perform information processing using electrical or optical spikes [25,26,27]. Various operational mechanisms, such as transistors, tunneling devices, and memristors, can be used for neuromorphic computing [28,29,30]. High-density integration with two-terminal memristors is emerging as a suitable approach for the fabrication of future devices characterized by low-power/low-thermal budgets and in-memory and on-chip computing [31,32,33]. A memristor combines the concepts of memory and resistors and exhibits resistive switching (RS) under an electrical bias resulting from the movement of anions and cations in materials. RS is generated by the formation of a conductive filament in the memristor [34,35,36,37,38,39,40,41,42]. When RS occurs from the high-resistance-state (HRS) to the low-resistance-state (LRS) owing to channel formation, the operation is called “SET”, whereas in reversed cases, it is termed “RESET” [43,44,45,46]. The RS operation mechanism can be classified depending on the materials, including transition metal dichalcogenides, transition metal oxides, boron nitride, silicon, or a layered combination of more than two materials [47,48,49,50,51,52,53,54,55,56]. A synapse array comprising memristors can provide synaptic characteristics and combine with a deep neural network for advanced data processing in neuromorphic computing.

This review focuses on recent research regarding hardware-implemented neuromorphic computing using memristor arrays, covering aspects ranging from operational mechanisms to intelligent applications. A neurobiological synapse-mimetic memristor array providing RS under electrical or optical spikes is reviewed, including the material candidates, synapse operation and characterization, and neuromorphic computing processes for practical applications, such as image sensors, pattern recognition, and image or pattern processing. Critically, AI-embedded chips comprising memristor arrays overcome the memory bottleneck in conventional von Neumann architectures. A resistive random-access memory (RRAM) operation can provide low-power intelligent data processing by mimicking neurobiological synapses and will play a crucial role in future AI and memory computing applications. Figure 1 compares the biological neuron system in the human brain with a memristor-based synapse array for human-brain-mimicking neuromorphic computing. In biological neurons, active potentials are created from the pre-synaptic to the post-synaptic neurons via Ca^2+^ channels in the synaptic cleft. The post-synaptic neuron absorbs the ion into the ion channel receptor and generates a neural signal to transfer it to the next neuron. This process occurs in the nervous system of the human brain, which is capable of cognitive thinking and object detection using optical nerves. Compared to biological neurons, memristors operate by forming conductive filaments under electrical spikes and can mimic synapses by acting as a large-scale crossbar array. By integrating memristor arrays with artificial neural networks (ANNs), hardware-embedded human-brain-mimicking neuromorphic computing can serve as an efficient platform for emerging technologies, such as those implemented in image processing, pattern recognition, the Internet of Things (IoT), and other AI tasks. Below, the technologies for memristor-based AI are categorized according to devices and basic operations, synaptic behaviors and synapse arrays, and convolutional neural networks (CNNs) or optic-integrated image sensors.

## 2. Single Memristor Device

A memristor is a two-terminal electrical component in which an active material is sandwiched between a top electrode (TE) and a bottom electrode (BE) [57,58,59,60]. Memristive behavior entails the functionalized hysteresis of electrical resistance and can be quantified by a current value corresponding to a voltage sweep [61,62,63,64]. As shown in Figure 1, a typical I–V curve of a memristor exhibits an identical “butterfly curve” shape by changing its resistance. When a positive voltage sweep is applied to the TE, the SET process occurs at voltages above a specific magnitude, leading to a significant current increase (Figure 1 (1)). After the setting process, the memristor sustains its HRS while the voltage decreases to zero (Figure 1 (2)). Notably, the current value is distinctly higher than that of the previous state at the same voltage [65]. A sequential negative voltage sweep occurs during the reset process, thereby changing the resistive state from the HRS to the LRS again over a specific negative voltage (Figure 1 (3)). Decreasing the negative voltage to zero completes one set and reset cycle, returning to the initial LRS (Figure 1 (4)). Although this I–V curve behavior differs depending on the mechanism, designed structure, current compliance, and voltage sweep range, the basic parameters of the memristive function are as follows.

(1)SET and RESET voltages

These indicate the voltages at which the set and reset processes occur, respectively. Smaller SET and RESET voltage ranges are considered favorable, as they lead to lower energy consumption. In some cases, the operating voltage range is deliberately adjusted to be lower than the SET and RESET voltage ranges to stabilize the overall operation [66,67].

(2)On/off ratio (HRS/LRS)

The on/off ratio is the HRS/LRS resistance ratio and a major performance-determining parameter of the memristor. The on/off ratio indicates the distinguishability between the set and idle states. An integrated RAM system using memristors with an insufficient on/off ratio suffers from scalability issues because the minimum read margin cannot be satisfied in a large array. In neuromorphic applications, a wide range of resistance states leads to large synaptic weight values, thereby facilitating the learning process [68,69].

(3)Stability

In a single-memristor device, continual set and reset processes must be demonstrated at a specific voltage. For example, residual cations trapped inside the active layer structure of memristors involving cation filament formation can hinder stable set and reset processes, as discussed in detail below. In RAM, the retention time of the resistance state is considered critical [70,71]. The cyclic stability of memristors is a persistent issue in practical computing applications. In addition to the aforementioned parameters, the imitability of synaptic and neuronal signal processing is another critical performance parameter in neuromorphic computing and is discussed below.

**Figure 1 sensors-23-03118-f001:**
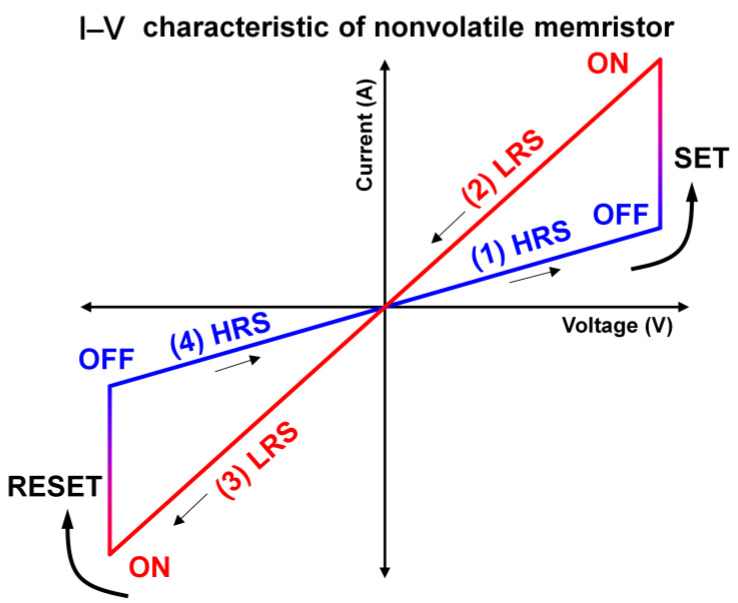
Typical I–V curve of a nonvolatile memristor under bipolar voltage sweep. The SET process occurs above a certain voltage threshold, and the resistance state changes from high-resistance state (HRS) to low-resistance state (LRS), thereby causing an abrupt increase in current (1). In (2), the device maintains the LRS; thus, the current value is considerably higher. During the negative voltage sweep, the device still retains the LRS (3); however, certain negative voltages lead to the RESET process, returning the LRS to the HRS (4).

An effective strategy for memristor operation entails the induction of dramatic changes in the resistance through the formation and rupture of ion filaments [72,73]. Under an applied electric field, ions migrate inside the host materials to form a filament, and when this filament grows and creates an ohmic contact between the TE and BE, the electrical resistance rapidly decreases (set process) [74,75]. Two types of mobile ionic species are primarily used in memristors: metal cations, such as those of Ag, Cu, and Ti, and anions, such as oxygen (or its vacancy) [76,77,78,79]. In the case of a metal cation film memory, metal ions are typically introduced into the host material from the electrode through an electroforming process, or the active layer itself is mixed in a stoichiometric ratio [80,81]. Various reported ion-mediated memristors and their rudimentary mechanisms are discussed below.

Figure 2a shows a memristor structure comprising silver metal cations as the filament material, as demonstrated by Yoon et al. [82]. When the formed filament is of sufficient thickness, it can persist for extended periods of time, even if the voltage applied after the set process returns to zero. Therefore, in the sequential negative voltage sweep, the LRS is maintained and the current is high (arrow 3 in Figure 1). Memristors that exhibit such memorizing behavior are referred to as “nonvolatile” types. In contrast, a cation filament of insufficient thickness maintains the set state only in the presence of applied voltage; when the voltage is lower than a specific value, the filament spontaneously disconnects and reset occurs. This phenomenon was investigated in detail by Wang et al., The dissolution behavior of the silver filament was explained by the Joule–Thompson effect, and the regaining behavior of the metal residue ion from the ruptured filament back into the bulk metal electrode occurred as a result of the Ostwald ripening phenomenon. Such memristors cannot “memorize” the resistance state and are therefore referred to as “volatile” memristors. Generally, volatile memristors are used as signal processors, which mimic neurons that determine thresholds and generate outputs only when the input exceeds the threshold. As the amount of silver is stoichiometrically controlled instead of using a bulk Ag electrode, this memristor exhibits volatile behavior, whereby a reset process occurs during a unipolar sweep. (Figure 2b) Another strategy for controlling the volatility of a metal ion filament memristor is by means of current compliance. Due to the limited current, the diameter of the formed filament is insufficient to maintain the resistance state, thereby exhibiting nonvolatile I–V properties. In this context, Yang et al., suggested tellurium as a unique cation mobile species, as tellurium shows a reverse property to the applied current compliance [83]. Owing to the low melting point of tellurium, filament formation thereof shows nonvolatile retention under a low current, whereas the filament ruptures readily under a high current. Based on this property, Yang et al., succeeded in fabricating a power-efficient cation filament memristor (in nonvolatile memorization, a lower voltage is preferred to acquire a higher readout margin). Another major mobile species is the vacancy of an anion such as oxygen. As shown in Figure 2c, an oxygen vacancy can also be utilized to form and rupture a filament, thereby changing the resistance. Generally, vacancy migration memristors can adopt oxide as an active layer, as its intrinsic structural vacancy can act as a mobile species while changing the valence locally [84]. Therefore, unlike cation filament memristors, anion filament memristors typically do not require the direct participation of an electrode; chemically inert electrode materials, such as Pt or Au, are primarily adopted. In addition to the manner in which the filaments are formed, memristive operation can be simply realized using the bulk migration of negative ions. Park et al. suggested a TiO_x_-based memristor that exhibits a gradual set process. They took advantage of the wide range of resistance states, thereby facilitating the emulation of neuromorphic applications. As shown in Figure 2d, when a positive voltage is applied on the TE, oxygen anions migrate to the TE, thereby rendering the BE region relatively more metallic and decreasing the thickness of the insulating TiO_x_ layer (Set, LRS) [85]. In contrast, a negative voltage causes reversed oxygen anion migration to the BE, which causes a transition to the HRS (Reset). Instead of the abrupt ohmic connection and disconnection occurring in the filament method, this bulk migration mechanism enables a more gradual set process (Figure 2e). For structures such as perovskites, both the cation filament and vacancy filament mechanisms can coexist in one device and successfully demonstrate memristive behavior. Recently, organic–inorganic halide perovskites have attracted considerable attention owing to their unique properties, such as tunable absorption, profuse ion migration, and mixed ionic–electronic conduction behavior; these are highly beneficial properties for memristor devices. Han et al., fabricated an RRAM device based on CsPbBr_3_ perovskite quantum dots (QDs) to study light-sensitive artificial synapses [86]. A QD layer was sandwiched between the two poly(methyl methacrylate) layers and deposited on an indium tin oxide (ITO)-coated polyethylene terephthalate (flexible) substrate, as depicted in Figure 2f. The RS behavior of this device was attributed to the formation and rupture of Br^-^ vacancies and metallic filaments (Ag), driven by an external bias voltage and photoirradiation, as shown in Figure 2g,h. The collective effects of electrochemical metallization (Ag filament formation) and valence change (Br-vacancy filament formation) in perovskite QD-based memory may contribute to elucidating the real logic circuit applications of perovskite materials. Moreover, as explained above, rather than using a single active layer, the construction of multiple active layers to achieve organized functionality and applying innovative designs can lead to improved memristor performance.

It is possible to enhance the capabilities of a memristor beyond a single switching layer by incorporating various other materials to form a heterostructure. Two examples of heterostructured architectures consisting of innovative materials that enable the implementation of their unique mechanisms are described below. Despite its fine integrity, the crossbar array suffers a fundamental limitation owing to the unwanted reverse current flow through the non-selected cells. This “sneak current” critically hinders the scalability of the memory array as it reduces the ratio of the actual signal from the selected cell (Figure 3a, red circle highlighted). As a novel solution, Sun et al., demonstrated a self-selective van der Waals heterostructured memristor composed of graphene and hexagonal boron nitride (h-BN) [87]. Unlike conventional solutions for sneak currents, this single device operates as a selector device; therefore, no additional complexity exists in terms of the circuit design. As depicted in Figure 3b, the sandwich-like structure comprising a graphene layer between two h-BN layers acts as a barrier to Ag ion diffusion while offering low electrical resistance. The diffusion barrier structure allows for two independent device operating mechanisms: boron vacancy (V_b_) filament formation and Ag filament formation (Figure 3c). Note that, despite the state of the nonvolatile V_b_ filament, the entire device experiences a sufficiently low current when the Ag filament is in the “off” state, regardless of whether the V_b_ filament is in the “on” state or not (Figure 3d, 1–2). Based on the high tunneling resistance of h-BN, the connection state of the Ag filament affects the device resistance dramatically (10^10^ ratio between LRS and unselected state, Figure 3e) such that negligibly small currents exist during read operations in the crossbar array. As the read, write, and reset voltages are higher than the set voltage of the bottom Ag filament, the sneak current from the unselected cells during a ½ V read is efficiently inhibited. The high selectivity allowed for crossbar array device scalability of up to 1 Tbit while satisfying the readout voltage margin criteria of 10% (in the worst-case scenario entailing 10 Ω wire resistance between cells, as shown in Figure 3f).

Sung et al., demonstrated a heterostructured memristor that simultaneously mimicked both neurons and synapses [64]. Figure 4a shows a structure in which a silver-filament-based memristor is combined with a phase-change memory (PCM) material using the formed silver filament as the BE, corresponding to the signal-processing roles of biological neurons and synapses, respectively. The fundamental switching principle dictating phase-change memristive synapses is the transition between amorphous and crystalline phases. This synapse comprises a Ge_2_Sb_2_Te_5_ PCM device, which is sandwiched between two electrodes and can be programmed based on the degree of phase change altered by electric-current-driven heat. Crystallization of the amorphous area occurs when the heat produced by the electrical current exceeds the crystallization temperature of the PCM. In contrast, when the temperature increases above the melting point, the crystalline region melts and transitions into an amorphous phase. Despite being commonly regarded as an energy-intensive device in comparison to other synaptic-switching devices, Sung and coworkers demonstrated that their distinct nano-filament heating structure generates a localized and intense electric field, resulting in energy-efficient phase changes in the top PCM layer [88]. Biological synapses regulate the strength of the signal connection between pre- and post-synapses based on the sensitivity and density of neurotransmitters. These connection strengths are called “synaptic weights”. While these synaptic weights remain nonvolatile, neuroplasticity is created, and memorization and learning take place. Similar to the plasticity of biological neurons, in which the synaptic weight increases as the time difference between input spike signals decreases, as shown in Figure 4b, the fabricated PCM layer exhibits synaptic-like nonvolatile connection strength storage behavior (Figure 4c). While biological synapses store connection strength information in a nonvolatile manner, biological neurons only play the role of firing action potentials, or output signals, to the next synapse when the incoming signal exceeds a threshold. In this “integrate-and-fire” process, the information from the former stimulation is considered volatile (Figure 4d). In the case of filament-forming memristors, this volatility can be realized as an unsustainable filament, that is, a volatile memristor (Figure 4e). Recent neurological studies have shown that neurons excited by a given stimulus have “intrinsic plasticity”, which plays an important role in the learning process. In particular, the length of the axon initial segment (AIS) increases, and its response of sending out action potential outputs becomes more frequent, as depicted in Figure 4f. As shown in Figure 4g, the relaxed state of the neuronal device shows tonic bursting spikes with relatively low frequency. However, under the same current input, sequential input causes the bottom threshold layer to “excite”, thus leading to more frequent tonic bursting. Although there is a dearth of explicit indications and material analyses regarding the presence of silver residues, this behavior is thought to arise from silver residues in the host material layer, emulating the AIS length change of biological neurons in signal processing. By combining a novel neuronal layer with a PCM synaptic device, researchers have created a single device capable of impelling the intrinsic plasticity of these neurons under low power at the device level, enabling a significantly enhanced learning performance. Thus, memristors are not limited to simple resistance changes and memory applications; they are expected to be critical and basic units of artificial synapses and neuro-mimetic devices. Depending on the variance in the electrical and optical spikes, artificial synaptic devices can be modulated using gradual conductance for multi-state device functionality. The salient concept of synaptic characteristics and the key role of memristors in modulating artificial synapses are discussed below.

## 3. Synaptic Characteristic Investigation in Artificial Synapse

(1)Biological synapses

The human brain is composed of neurons, that is, nerve cells that communicate with one another via synapses. Synapses are located between the axon end of the pre-synaptic neuron and the dendrite of the post-synaptic neuron, as shown in Figure 5a [89]. When a neuron receives an electrical signal, it conveys it to the post-neuron via synapses by generating an electrical spike, known as an action potential (spike). When an action potential is generated at the axon end of the pre-synaptic neuron, neurotransmitters are released through the synaptic cleft, as shown in Figure 5b. Synapses are small spaces (20–40 nm) between the axon end of the preceding neuron and the dendrites of the next neuron. Next, the chemical signal (neurotransmitter) is transmitted to the dendrite of the post-synaptic neuron. The capacity of synapses for the strengthening or weakening of chemical signals depending on the mobility and concentration of transmitters is referred to as “synaptic plasticity” [90]. Essentially, synaptic plasticity is a type of brain activity occurring in the synapses that serves to modify the connection strength. Synaptic plasticity is important in learning and memory because it is associated with the creation of short- and long-term memories. It is based on the number of synapses and neurotransmitters and the effectiveness of cellular actions [91].

(2)Artificial synapses

Synaptic plasticity in biological synapses depends predominantly on the connection strength between neurons; it can become stronger (potentiation) or weaker (depression) depending on the synaptic weight. Artificial synapses are composed of a TE that mimics a pre-neuron, a BE as the post-neuron, and an active layer sandwiched between them that functions as a biological synapse, as shown in Figure 5c. The conductance and resistance of the devices govern the strength of the connection between the pre- and post-neuron, representing the synaptic weight. Artificial synaptic memory is categorized into two types depending on the retention time of the derived memory: short-term plasticity (STP) and long-term plasticity (LTP) [92,93]. In addition, plasticity can be further categorized into spike-time-dependent plasticity (STDP) and spike-rate-dependent plasticity (SRDP) [94].

(3)Short-term plasticity (STP) and long-term plasticity (LTP)

Potentiation and depression are synaptic weight characteristics in biology; a positive value represents potentiation, and a negative value indicates depression. STP is the synaptic plasticity associated with short-term memory. STP is governed by the temporary potentiation and depression of the synaptic weight, as shown in Figure 5d (blue line). STP is beneficial for the performance of critical computational functions in spatiotemporal information processing. In contrast, LTP is the synaptic plasticity associated with long-term memories (red line), requiring long-term potentiation for learning and memory functions in synaptic devices [95]. LTP can be further classified into long-term potentiation and depression, resulting from the strengthening and weakening of the synaptic weight, respectively. In addition, in biological synapses, plasticity can be transformed from STP to LTP through repeated pulse application, which increases the synaptic strength significantly. In 2011, Onho et al., systematically investigated the influence of the input pulse quantity on the generation of an Ag filament in an Ag_2_S device and its transition from STP to LTP [96]. In general, STP can be classified into paired pulse facilitations (PPFs) and paired pulse depressions (PPDs), as depicted in Figure 5e. When the series of transmitted spike signals is sufficient, PPFs increase in STP, whereas PPDs decrease. Currently, neuromorphic systems are implemented by combining a number of volatile STP and nonvolatile LTP components [53,97].

(4)Spike time-dependent plasticity (STPD) and spike rate-dependent plasticity (SRPD)

In ANNs, the parameters affecting synaptic plasticity must be studied as they govern changes in the connection strength. In 1949, Hebb et al., suggested a basic mechanism for learning and memory modeling by noting that the synaptic strength increases when the activity in the pre-synaptic neuron repeatedly induces action potentials in the post-synaptic neuron, as shown in Figure 5f [95,98]. The above-described Hebbian learning concept can be further explained using two factors: STDP and SRDP. STDP defines synaptic strength as a function of the timing between the pre-synaptic and post-synaptic action potentials. In STDP, the change in the synaptic weight (ΔW) depends on the relative timing between the pre- and post-synaptic pulses, as demonstrated in Figure 5g. If we consider t_pre_ and t_post_ as the arrival times of the pre- and post-synaptic pulses, respectively, the synaptic weight increases or decreases according to the time difference (Δt = t_pre_ – t_post_) between pre- and post-synaptic neuron activity [99]. The synaptic change is represented by Δt; if Δt is small, ΔW is large, and vice versa. Another essential concept of synapses is the SRDP, where ΔW is dependent on the firing frequency, as shown in Figure 5h [100]. Here, synaptic STP and LTP occur under stimulation by low- and high-frequency spikes, respectively. In general, pre-synaptic spikes with a high frequency ˃10 Hz lead to potentiation, whereas pre-synaptic spikes with a low frequency ˂10 Hz lead to depression [101]. The following section summarizes novel and creative approaches that effectively mimic the signal-processing characteristics of the above-mentioned synaptic properties in terms of their structure and architecture.

**Figure 5 sensors-23-03118-f005:**
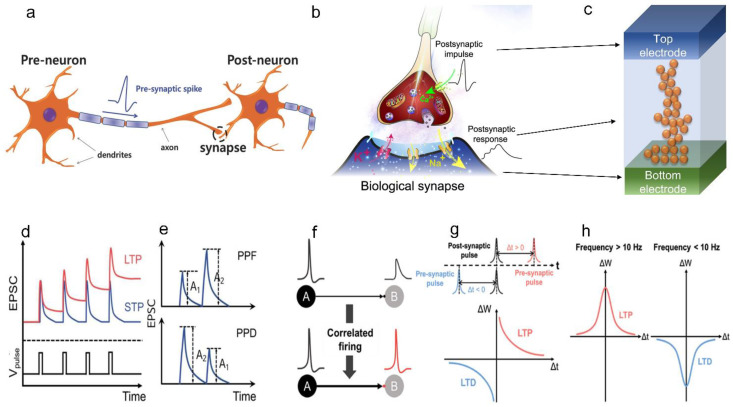
Comparison between biological and artificial synapses. (**a**) Biological synapse, pre-neuron, and post-neuron; adapted with permission from ref. [89], copyright 2016 Wiley-VCH Verlag GmbH & Co. (**b**) Schematic of an artificial synapse; adapted with permission from ref. [77], copyright 2021 Springer Nature. (**c**) General model of a two-terminal memristor-based artificial synapse. (**d**) Long-term potentiation (LTP) and short-term potentiation (STP) of artificial synapses are shown in red and blue, respectively. (**e**) Paired pulse facilitation (PPF) and paired pulse depression (PPD) in artificial synapses. (**f**) Schematic of the Hebbian learning model. (**g**) Spike time-dependent plasticity (STDP) according to the relative timing of the pre- and post-synaptic pulses. (**h**) Spike rate-dependent plasticity (STRP) for long-term potentiation (LTP) and long-term depression (LTD) occurring under distinct conditions; adapted with permission from ref. [101], copyright 2021 Wiley-VCH Verlag GmbH & Co.

Li et al., suggested a temporally and spatially complete neural network system derived from dendrites with nonlinear integration and filtering [102]. The designed ANN comprehensively mimics the biological neuron system comprised of synapses, dendrites, and soma using memristor devices that can perform a digit-recognition task by stimulating the multi-layer network. The biological neuron system and the ANN are compared in Figure 6a. The synapses, dendrites, and somas in the biological neural network correspond to plasticity/weight, integration/filter, and integration/fire in ANN systems for computing functions, respectively. Figure 6b illustrates the architecture of an ANN, composed of input, hidden, and output layers behaving as dendrites, and a soma, showing nonlinear integration and filtering functions. To better understand the operational mechanisms, a biological neural system (Figure 6c) and an ANN (Figure 6d) are compared. The membrane of biological dendrites controls the movement of ions (Na^+^, Ca^2+^, and Mg^2+^) via *N*-methyl-d-aspartate channels (Figure 6c). Similarly, conventional metal-oxide-based memristors operate under an electrical potential created by ion migration within the material, thereby mimicking dendrites (Figure 6d). The electrical response of an artificial dendrite exhibits nonlinearity under a gradually applied voltage (Figure 6e) and nonlinear filtering and integration properties (Figure 6f). A hardware-embedded ANN with artificial dendrites and a soma has been demonstrated, as shown in Figure 6g–i. A schematic of an artificial synapse array comprising 1024 nonvolatile memristors with one-transistor–one-resistor configurations is shown in Figure 6g. An artificial neuron consisting of artificial dendrite and soma devices is shown in Figure 6h. The equivalent circuit diagram is depicted in Figure 6i, showing the bridging of the synapse array and soma by the dendrites. Figure 6j shows the firing spikes measured during the inference process of an ANN containing artificial dendrites (left) and without dendrites (right). In the case without dendrites, the contrast in the number of spikes between the matched and unmatched cases is smaller than that with dendrites. The number of measured spikes in the matched cases is greater than that in the unmatched cases when operating with dendrites. The functionality of filtering and integration with dendrites and soma provides a fully mimetic ANN with the advantages of energy efficiency and accuracy for complex AI tasks.

## 4. Fully Imitated Multi-Sensory Perception

Wang et al., demonstrated an artificial lobula giant movement detector (LGMD) using a light-mediated few-layer black phosphorous nanosheet-CsPbBr_3_ perovskite QD (FLBP-CsPbBr_3_) threshold switching (TS) memristor for movement and light-sensitive detection [103]. Figure 7a shows a schematic of the biological apposition compound eye of a locust, comprising the corneal facet, screening pigment, and rhabdom under an optic signal. Owing to its unique appositional compound eye, the locust has a large field of view (FoV) and angular-sensitive omnidirectional sensing. To mimic the large-FoV detection capability, a hemispherical artificial biomimetic compound eye was fabricated using 20 × 20 FLBP-CsPbBr_3_ TS memristor flexible arrays on a retinal PDMS shape, as shown in Figure 7b. The incident-angle-dependent photocurrent was measured along the x direction, exhibiting the highest photocurrent at 90° (Figure 7c) for the optical pulse (365 nm wavelength, 0.72 mW power, and 10 ms pulse width). Figure 7d shows a schematic of the artificial synaptic behavior under light stimulus and the electronic pulses of the designed FLBP-CsPbBr_3_ TS memristor. The collision avoidance response in the artificial biomimetic compound eye is characterized by a voltage stimulus (+0.2 V pulse, 50 μs interval, and 50 μs duration). The corresponding conductance modulation of the FLBP-CsPbBr_3_ TS memristor was measured (Figure 7e), showing excitatory and inhibitory responses, which emulate biological LGMDs. This synaptic response can mimic collision predictions from looming objects before impact. Practical collision avoidance functions have already been attained using energy-efficient biomimetic designs by combining artificial TS memristors with dynamic robot applications.

Zhu et al., developed a multi-mode-fused spiking neuron (MFSN) for mimicking a somatosensory system to enable fully imitated multi-sensory perception [104]. The MFSN comprised a pressure sensor and NbO_x_-based memristor for temperature sensing. By decoupling the output signals according to the frequencies and amplitudes, simultaneous sensing of pressures and temperatures from fused spikes was demonstrated using the MFSN. Figure 7f compares the operational mechanisms of a biological somatosensory system with those of the artificial somatosensory system. The human neural system via which multiple stimuli (temperature, weight, and shape) are sensed can be classified based on the biological somatosensory system and the presence of thermal receptors and mechanoreceptors. Signals are delivered to the cerebral cortex by various spikes, such as pressure, multi-mode-fused, and thermal spikes. Similarly, the artificial somatosensory system based on MFSN arrays comprising a pressure sensor and NbO_x_-based memristor device can simultaneously detect pressure and temperature stimuli. As an output, various frequencies and amplitudes are generated for the pressure, multi-mode-fused, and temperature spikes. For practical applications using the MFSN array, a spiking neural network classifier is utilized for processing the multi-modal sensory inputs. The MFSN includes a piezoresistive pressure sensor and NbO_x_-based, temperature-dependent TS memristor. Depending on the applied pressure and temperature changes, it generates output spikes with distinct frequencies and amplitudes. The working principle of the NbO_x_-based TS memristor in the MFSN is shown in Figure 7g. The crossbar of the Ti/Pt/NbO_x_/Ti/Pt structure is utilized for the memristor device and exhibits a stable I–V sweep over 50 cycles. At the threshold voltage (“V_TH_”) of ≈1.34 V, a sudden conductance change is observed (SET) for the LRS. During the reverse sweep, the current is reduced at the hold voltage (“V_H_”) of ≈1.04 V; the MSFN switches to the HRS (showing volatile switching behavior). The core-shell model of the NbO_x_-based memristor for temperature-dependent TS behavior is designed using a corresponding simulation program with an integrated circuit emphasis model (Figure 7h). The NbO_x_-based operation can be explained based on the substoichiometric Nb_2_O_5-x_ region (shell) as a conductivity–temperature-dependent semiconductor and filamentary core region of the NbO_2_ channel. The fabricated 3 × 3 MFSN array is utilized as a spiking neural network for pattern recognition and classifying objects with various shapes, temperatures, and weights for practical applications. Based on the aforementioned biomimetic array structures, higher-level cognitive functions for practical applications, such as image processing and pattern recognition, are discussed below. We summarizes the various memristive device and their characteristic as shown in Table 1.

## 5. Multi-Layer Perceptron for Pattern Recognition

Synapses can be trained by neural networks for emerging future technologies, such as in edge computing, image and pattern recognition, and neuromorphic computing; for example, AI hardware [48,121,122,123,124]. Using the dataset in the Modified National Institute of Standards and Technology (MNIST) database, handwritten using a multi-layer perceptron (MLP), simple recognition of patterned numbers by synaptic devices has been demonstrated [121,125,126,127]. Choi et al., designed dislocated SiGe layers with cracks propagated through the films arising from the lattice mismatch between Si and Ge (Figure 8a) [128]. This structure functioned as an Ag ion pathway for reliable neuromorphic computing. Filament formation through the pre-designed dislocated pathway enabled reproducible RS with minimal variation. As another example, an epitaxially grown SiGe-based random-access memory (epiRAM) exhibited an enhanced RS after the threading dislocation was widened using the Schimmel etch method (Figure 8b–d). EpiRAM exhibits a temporal variation of 1% after 700 I–V sweep cycles. Depending on the etching time for dislocation widening, the set voltage is reduced owing to enhanced filament formation while the set voltage variation increases. The Ge content and epiRAM size did not result in significant differences; thus, only filament formation at the pre-desired dislocated area participates in RS. The RS and reproducible characteristics of epiRAM are utilized for supervised learning using the MNIST handwritten recognition data set (Figure 8e–g). The simulation (Figure 8e) uses a stochastic gradient descent weight update with the MLP algorithm. A three-layer neural network comprises 28 × 28 input neurons (MNIST images), 300 hidden neurons, and 10 output neurons (classes of 0–9 digits). The 300 hidden neurons comprise the summation from the input neuron, activation, and binarization. Figure 8f illustrates the epiRAM crossbar array, functioning as a synapse layer with the peripheral circuit. Pattern recognition accuracy was simulated as shown in Figure 8g for both ideal software (blue) and epiRAM (red). In particular, 10,000 patterns were selected as a test set after being pre-trained by 60,000 patterns. epiRAM exhibited 95.1% accuracy for pattern recognition, whereas the ideal software exhibited 97%. Thus, strategically engineered dislocation demonstrated unprecedented reproducible neuromorphic computing by providing pre-desired filament pathway-based RS based on nonvolatile memory.

Seo et al., fabricated an optic-neural synaptic (ONS) device by combining synaptic (WSe_2_/weight control layer (WCL)/h-BN) and optical (WSe_2_/h-BN) sensing functions capable of both colored and color-mixed pattern recognition [129]. Schematics of the human eye and the visual-cortex-mimicking artificial ONS device are shown in Figure 8h. Depending on the incident wavelength, the resistance of the WSe_2_/h-BN photodetector (PD) changes. The generated carriers in the reduced-resistance state in the WSe_2_ induce an increase in trapped carriers in the WCL. This modulates the synaptic operation of the ONS depending on the incident light wavelength. To confirm the wavelength-dependent synaptic dynamics of the ONS, they studied the post-synaptic current and LTP/LTD for the synaptic plasticity of the ONS under three wavelength conditions: red (R, λ = 655 nm), green (G, λ = 532 nm), and blue (B, λ = 405 nm). The WCL was formed by O_2_ plasma treatment on the h-BN films, producing oxidized boron. The synaptic cleft terminal (Figure 8i) generates a hysteresis characteristic between the pre-synaptic and post-synaptic terminals depending on the applied voltage in the synaptic cleft terminal. Using the ONS devices, an artificial optical neural network (ONN) was developed for colored and color-mixed pattern recognition based on a perceptron network model. As shown in Figure 8j, a CNN (left) executes color-filtering using a neuron array, whereas the ONN (right) has an optical recognition function related to the synapse. As an input signal, the voltage is determined depending on the color (1 V for R, 0.5 V for G, and 0.3 V for B); three neurons and a 28 × 28 array group are used for each cone cell group. Each cone cell group is fully connected to classifying neurons (“1” and “4”) and generates six classifying neurons as an output. In their study, the MNIST dataset was selected for the pattern recognition task (image size is 28 × 28) with modification for color-mixed patterns. They prepared six types of training datasets with 100 images in each dataset and nine types of test datasets with 20 images for each dataset. The 600 training images were used for the pattern recognition task, as shown in Figure 8k, and a high recognition rate (>90 %) was attained by the ONN (red) after the 50th epoch; in contrast, the CNN (black) exhibited a low recognition rate (<40%). As the training epoch increased, the recognition rate for the mixed-color pattern was optimized in the case of the ONN. The weight mapping of the 12th (left) and 600th (right) training epochs confirms the successful gradual weight optimization during training (Figure 8l). After training, the blue-colored “4” was successfully recognized (activation of ‘B4′ output neuron) by both the CNN and ONN, as shown in Figure 8m. However, in the case of a mixed-color number (red and green), the highest activation value was at “R1” for the ONN and “G4” for the CNN; thus, the ONN demonstrated successful color recognition (Figure 8n).

## 6. Convolutional Neural Network

The recognition or processing of actual images using a CNN is an essential component of neuromorphic computing. Memristor crossbar arrays or synaptic device arrays enable nonvolatile memory applications specially designed for signal and image processing integrated with neural networks [93,130,131,132,133]. The Canadian Institute for Advanced Research (CIFAR)-10 dataset is typically used to evaluate recognition task performance [134,135,136]. The addition of image processing kernels to neural networks and arbitrary processing of input images can also indicate the feasibility of next-generation neuromorphic-based image processing and edge computing [137,138,139,140,141].

Seo et al., developed an optogenetics-inspired optoelectronic synapse using layered rhenium disulfide (ReS_2_) with sulfur vacancies (S_v_) [142]. Harnessing the intrinsic persistent photoconductivity (PPC) effect, optogenetic activity was used to mimic the neural system of a biological synapse. The conductivity of ReS_2_ was modulated under the incident optical stimulation, and photosensitive memory was created using the PPC effect. Using the optogenetics-inspired ReS_2_ array, the authors demonstrated a hardware-based neural network (HW-NN) through a CIFAR-10 dataset recognition task (Figure 9a). The CIFAR-10 dataset was examined using an eight-layer CNN comprising an ReS_2_ synapse array integrated with deep neural network+ NeuroSim. The CIFAR-10 dataset comprises three color channels (red, green, and blue), ten categories of objects, 50,000 images for the training set, and 10,000 images for the inference set. In the feed-forward CNN operation, the training input (CIFAR-10) signals are processed through the six convolutional layers (layers 1–6), followed by the fully connected layers (layers seven and eight). Layers two, four, and six function as pooling layers for reducing the feature size. The input pixels corresponding to the voltage (V) are convoluted in multiple kernel layers using synaptic weights (W). During the convolution operation (I = W × V), the current signals are calculated and processed by rectified linear unit activation functions, resulting in voltage signals (V = f_RELU_(I)) for the next convolution layer. In layer six, a flattening operation is conducted, and the voltage signal is transferred to the fully connected layers (layers seven and eight). Figure 9b shows a schematic of the ReS_2_ synapse-based HW-NN, illustrating the operation principle and peripheral circuits. The conductance difference (weight) is calculated for two synapse devices (potentiation and depression). Details of the weight calculation and update via light stimulation in the optogenetic-inspired optoelectronic synapse are shown in Figure 9c. The recognition rate for the CIFAR-10 dataset was confirmed to be 89.4% by the ReS_2_-based optoelectronic synapse and 91.1% by the ideal synaptic devices (Figure 9d). Even after the ReS_2_ synapse underwent 1000 cycles of bending, 89.2% of the recognition rate was retained.

Yeon et al., demonstrated an Ag–Cu alloying conducting channel in an Si memristor, achieving reliable neuromorphic computing for data retention and image processing (Figure 9e–k) [143]. The silicidable copper stabilizes Ag ion migration during filament formation and rupture. In addition to the Ag–Cu alloy, they suggest other possible channel combinations (Ag–Ti, Ag–Cr, and Ag–Ni) for the TE. Cu, Ni, Cr, and Ti are thermodynamically stable and interact with Si, consequently resulting in the formation of a stable interface between the conduction channel and Si; however, these metals do not exhibit RS characteristics. In contrast, because Ag is thermodynamically unstable in Si, the resultant electrochemical mobility induces memristive behavior. Accordingly, the utilization of a binary electrode (Ag with a silicidable metal) delivers a reliable memristor for neuromorphic computing. Ti is miscible with Ag but exhibits a lower diffusivity than Ag in Si media and cannot guarantee scaffold formation before Ag migration. Cr and Ni exhibit higher diffusivities than Ag in Si but are immiscible with Ag; thus, they exhibit superior memristive switching uniformity but poor long-term stability. However, the diffusivity of Cu is higher than that of Ag in Si (favorable backbone formation) and it is partially miscible with Ag (it stabilizes the Ag ions in Si via bridging). The designed Ag–Cu memristor array was fabricated for a 32 × 32 transistor-less Si memristor; Figure 9e–g shows a digital photograph, optical microscopy image, and scanning electron microscopy image of the array, respectively. To evaluate the stability of the Ag–Cu memristor array, its data retention capability was tested, as shown in Figure 9h (initial state (left) and after 60 s (right)). It successfully maintained a 256-level greyscale image during the 60 s. However, in the case of Ag–Ni and Ag, the image information could not be stored, and poor retention was observed. These results correspond to the stability of the memristor, controlled by the alloying conducting channel. For functional tasks using the Ag–Cu memristor array, convolutional kernels are integrated for image processing (Figure 9i–k). Four types of kernels (sharpened, softened, and edges (horizontal and vertical)) were programmed in the convolutional process for each pixel. The image processing kernels were programmed into four columns of the Ag–Cu array and processed in parallel. Differential pairs (two memristors in the same output column) receive either positive or negative input pixels (Figure 9i). The four types of image processing kernels are shown in Figure 9j. Based on the kernel operation, the processed image shows four types of well-processed results relative to the input image (Figure 9k).

## 7. Light-Sensitive Synaptic Device for Image Sensor

The human eye and visual cortex mimetic chips are the most powerful devices for AI-embedded hardware-based neuromorphic computing [144,145,146,147]. Real-time weight updates from input signals provided by light-based patterned images, as performed by the human eye, and signal processing in a neural network (human brain) need to be developed for practical applications [148,149,150,151]. Although AI chips composed of light sensors and neural systems still face challenges emulating biological neurons in the human eye and brain, several state-of-the-art methods have been developed for the integration of desired components [152,153,154,155,156].

Choi et al., developed heterointegrated chips comprising optoelectronic devices (light-emitting diode (LED) and PD arrays) and neuromorphic cores (memristor crossbar arrays) [157]. In their system, stackable and replaceable chips were embedded for classifying light-based input image information. Data processed in parallel using these reconfigurable AI-embedded chips provided enhanced data bandwidth and energy-efficient edge computing. Heterointegrated processing modules (image sensors, kernel processors, and noise-reducing signal processors) were assembled for the recognition of light-based input patterns by the chip layers containing a denoising processor. Figure 10a depicts the schematics of reconfigurable and stackable heterointegrated chips. Light-based letter images (“M”, “I”, and “T”) are captured by PDs and LEDs in the eye layer. The stackable heterointegrated chips and chip-to-chip communications from the sensory input to the output, AI tasks, such as letter recognition or object detection, are illustrated in Figure 10b. To enhance the recognition performance, a denoising layer comprising a neural network can be inserted, as shown in Figure 10c. A 25 × 5 × 25 denoising autoencoder is used to denoise the corrupted images (5 × 5 patterned images). After denoising the corrupted image (Figure 10c), the letter recognition performance was enhanced, particularly between the letters “I” and “T” (Figure 10d). The denoised “T” kernel exhibits a higher current difference than in the case without denoising. Thus, the heterointegrated AI-embedded chip enables highly versatile data processing via neuromorphic computing.

Zhou et al., developed Pd/MoO_x_/ITO optoelectronic resistive random-access memory (ORRAM) based on ultraviolet (UV)-light-triggered nonvolatile and volatile resistance switching under tunable light intensity (Figure 10e) [158]. Under UV illumination, the oxidation state of Mo changes from Mo^6+^ to Mo^5+^, thereby resulting in a conductance variance from the HRS to the LRS when operating as an ORRAM. Photogenerated protons (H^+^) form hydrogen molybdenum bronze (H_y_MoO_x_) to induce an optoelectrical SET process, whereas the protons drift toward the Pd electrode under an electrical field for the RESET process. The memristive characteristics of ORRAM were investigated as shown in Figure 10f. Before UV illumination, ORRAM exhibited the HRS state (black line), switching to the LRS under UV illumination (blue line). During the RESET process, the electrical-field-induced proton drift to the Pd electrode produced an intrinsic resistance change (red line). Figure 10g compares the visual information transmission of the human optic nerve (upper part) with that of ORRAM-based neuromorphic computing (lower part). In the human eye, the information detected in the retina is transferred by the optic nerve to the visual cortex for recognition. The ORRAM mimics the sensing and pre-processing of the human eye and retina (neuromorphic pre-processing) and the output information is transferred to a three-layer ANN for image recognition. The role of pre-processing in the ORRAM is evaluated based on patterned letter images (“P”, “U”, and “C”) to highlight the processed noise, which is compared before (left) and after (right) pre-processing in the pattern (Figure 10h). In addition, the recognition rates with (black) and without (red) the ORRAM are compared in Figure 10i. In the case of the ORRAM, the recognition rate reaches 0.986 within 1000 training epochs, whereas 2000 training epochs were required for the same rate without the ORRAM. Under optical stimulation, the tunable synaptic plasticity results in human-eye-like signal sensing and neuromorphic computing, allowing for an efficient reduction in the data processing energy for visual information retention and recognition.

## 8. Conclusions

Hardware-implemented neuromorphic computing based on memristor arrays was reviewed herein, including the operation mechanisms of single memristors as well as those of crossbar arrays for intelligent applications, such as pattern recognition, image processing, and AI chips. Moore’s law faces the challenge of attaining a decreasing physical scale in semiconductor processing while enhancing device performance. Emerging human-brain-inspired neuro-morphic computing aims to address the memory bottleneck associated with von Neumann architectures, which hinders memory storage and processing in big data and AI areas. Memristor-based RRAM is emerging as a promising candidate for overcoming the memory bottleneck, as it allows for high-density integration and energy-efficient memory processing using neuromorphic computing. Beyond the two-terminal memristor array, to satisfy multi-bit data storage, heterosynaptic plasticity is desired for tunable synaptic function, similar to that operating in the human brain. Although the memristor-based one-resistor (1R) RRAM structure is simple and capable of high-density integration, its synaptic modulation performance is poor. Therefore, one-transistor–one-resistor (1T1R) arrays should be introduced for complex AI computing and data processing tasks. Diffusive metal electrodes, such as Ag, Cu, and Ti, degrade memristive switching owing to residue filament formation in the channel. Even when external-substance-induced memristive switching occurs, the degraded endurance of RRAM critically limits practical applications. Thus, in addition to cation filament formation, vacancy migration or tunneling-based synaptic devices, which generate memristive switching inside the channel, also display significant potential for reliable neuromorphic computing. To mimic the human nerve system with memory applications, it is essential to understand the specific functionalities and characteristics of the nervous system, such as those of the soma, dendrites, and nodes of Ranvier. Using these components, advanced neuromorphic computing based on realistic bio-inspired artificial neural systems can be developed. Furthermore, integration with optic functionality broadens the potential applicability to include optogenetic and photo-induced memory. The fabrication of 2D-material-based large-scale RRAM arrays remains challenging owing to limited synthetic methods and low-yield processing. To demonstrate high-performance neuromorphic computing, fundamental advanced synthetic approaches that deliver high-uniformity in wafer-scale should be considered. The abovementioned strategies, namely, vacancy-induced migration, realistic nervous system imitation, and optogenetic integration, improve the functionality and endurance of present neuromorphic computing. Memristor-based neuromorphic computing offers significant potential and functionality for in-memory processing and edge computing and should therefore be investigated further for future AI technologies.

## Data Availability

Not applicable.

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
