# Peer review of "Synapse-Mimetic Hardware-Implemented Resistive Random-Access Memory for Artificial Neural Network"

_sensors, 2023, doi:10.3390/s23063118_

Round 1

Reviewer 1 Report

I have read manuscript entitled “Synapse-Mimetic Hardware-Implemented Resistive Random-Access Memory for Artificial Neural Network” in a fairly detailed fashion. This review paper summarizes the results related to hardware-implemented neuromorphic computing based on a memristor array. In general, the topic of this manuscript is academically and technologically relevant and I think that manuscript in this version can be accepted for publication in Sensors. A careful revision of English is recommended.

Author Response

We thank the acceptance. we conducted additional English proofreading by a professional company (Editage Co.).

Reviewer 2 Report

In their manuscript the authors review the state of the art about the use of RRAM devices for bio-inspired neuromorphic computing applications. Overall, the review is sufficiently broad and covers different RRAM technologies, describes different bioinspired characteristics, and describes different in-memory computing hardware solutions based on these concept for sensor processing applications.

Here follows a couple of comments for the authors:

11) It would be helpful for the readers to include a paragraph and or a table that summarizes the desired ideal device performance that should be targeted in the future for practical application and a possible introduction of the described technologies on the market, and a comparison with the performance that can be achieved with current devices. Also, a discussion of the current key technological challenges that the researchers should focus on, would help the readers to better understand the state of the arts.

22) The English writing requires some improvement as some sentences are hard to read and some terms seems not correct, for instance, in the first paragraph in page 2, “… the constant data processing between the memory and the processor”, should likely be “… the constant data transfer between the memory and the processor”.

Author Response

#Reviewer 2

In their manuscript the authors review the state of the art about the use of RRAM devices for bio-inspired neuromorphic computing applications. Overall, the review is sufficiently broad and covers different RRAM technologies, describes different bioinspired characteristics, and describes different in-memory computing hardware solutions based on these concept for sensor processing applications.

Here follows a couple of comments for the authors:

  • It would be helpful for the readers to include a paragraph and or a table that summarizes the desired ideal device performance that should be targeted in the future for practical application and a possible introduction of the described technologies on the market, and a comparison with the performance that can be achieved with current devices. Also, a discussion of the current key technological challenges that the researchers should focus on, would help the readers to better understand the state of the arts.

We thank the reviewer’s comments. According to the reviewer’s comments, we summarizes the various memristive device and their characteristic as shown in Table 1. Also, in conclusion, we discussed the current key technological challenges and perspective on neuromorphic computing research.

(2) The English writing requires some improvement as some sentences are hard to read and some terms seems not correct, for instance, in the first paragraph in page 2, “… the constant data processing between the memory and the processor”, should likely be “… the constant data transfer between the memory and the processor”.

We thank the reviewer’s comments. According to the reviewer’s comments, we revise the sentence and conduct additional English proofreading by a professional company (Editage Co.).

Reviewer 3 Report

In this review manuscript, the authors have reviewed the switching behavior of resistive random-access memory, analyzed the resistive mechanism and performance requirements, and extended the merits of individual devices to neuromorphic applications, such as image processing and pattern recognition. The authors have done a lot of work, investigating recent demonstrations of neuromorphic computation using memristors and with an easy-to-understand presentation. However, there are some issues need to be addressed. The specific comments for the manuscript are listed below:

(1)    The description in the abstract is confusing, “Convolutional neural networks (CNNs), …, represent a breakthrough in artificial intelligence owing to their higher performance and lower power consumption compared with von Neumann architectures.” CNN as a model for software simulation which should not be contrasted with computer hardware architecture - such a representation is inaccurate.

(2)    The main drawback of this review is that the authors list various two-dimensional materials to achieve human-like brain functions, but lack analysis and critical thinking about the shortcomings of the various materials. For example, phase change materials are based on Joule heat and have a high energy loss (second paragraph on page 8). The integrated process of perovskite materials is toxic to Pb ions and not environmentally friendly (first paragraph on page 13), etc.

(3)    The authors should strengthen the logical character of the review. For example, after the first paragraph on page 8 introduces a high switching ratio self-selecting device to solve the leakage current problem, yet the second paragraph introduces a heterogeneous device with phase change and volatility to simulate neurons and synapses. Similar problems occur in other sections.

(4)    The authors should have supplemented with the opportunities and challenges of taking synaptic materials to applications at the end of the article, rather than simply piling up and listing the publications, a critical appraisal of the existing research should be made.

Author Response

#Reviewer 3

In this review manuscript, the authors have reviewed the switching behavior of resistive random-access memory, analyzed the resistive mechanism and performance requirements, and extended the merits of individual devices to neuromorphic applications, such as image processing and pattern recognition. The authors have done a lot of work, investigating recent demonstrations of neuromorphic computation using memristors and with an easy-to-understand presentation. However, there are some issues need to be addressed. The specific comments for the manuscript are listed below:

  • The description in the abstract is confusing, “Convolutional neural networks (CNNs), …, represent a breakthrough in artificial intelligence owing to their higher performance and lower power consumption compared with von Neumann architectures.” CNN as a model for software simulation which should not be contrasted with computer hardware architecture - such a representation is inaccurate.

We thank the reviewer’s comments. According to the reviewer’s comments, we revise the abstract as shown below. We changed the sentence as shown below.

→ (Page 1) “Neuromorphic computing, the most powerful artificial networks for complicated image processing and recognition, represent a breakthrough in artificial intelligence owing to their enhanced performance and lower power consumption compared with von Neumann architectures.”

  • The main drawback of this review is that the authors list various two-dimensional materials to achieve human-like brain functions, but lack analysis and critical thinking about the shortcomings of the various materials. For example, phase change materials are based on Joule heat and have a high energy loss (second paragraph on page 8). The integrated process of perovskite materials is toxic to Pb ions and not environmentally friendly (first paragraph on page 13), etc.

We thank the reviewer’s comments. According to the reviewer’s comments, we revise the sentence in revised manuscript.

(Page 9) “Despite being commonly regarded as an energy-intensive device in comparison to other synaptic switching devices, Sung and coworkers demonstrated that their distinct nano-filament heating structure generates a localized and intense electric field, resulting in energy-efficient phase changes in the top PCM layer.”

And we also pointed out that there is weak logical connection between silver residue and intrinsic plasticity, due to lack of direct evidence such as material evaluations.

(Page 9) “Although there is a dearth of explicit indications and material analyses regarding the presence of silver residues.”

(Page 14) “With the advantage of CsPbBr3 material, there are some toxicity issues on Pb ion which can hinder human-friendly applications such as flexible skin-like electronics and wearable sensor.”

  • The authors should strengthen the logical character of the review. For example, after the first paragraph on page 8 introduces a high switching ratio self-selecting device to solve the leakage current problem, yet the second paragraph introduces a heterogeneous device with phase change and volatility to simulate neurons and synapses. Similar problems occur in other sections.

We thank for reviewer’s advice. Thus, we revised with addition of sentences that can logically link between paragraphs.

(Page 6) “Various reported ion-mediated memristors and their rudimentary mechanisms are dis-cussed below.” for readers to understand following review of various cases of memristors.

(Page 7) “It is possible to enhance the capabilities of a memristor beyond a single switching layer by incorporating various other materials to form a heterostructure. Two examples of hetero-structured architectures consisting of innovative materials that enable the implementation of their unique mechanisms are described below.” for introducing two memristors with totally different structures, before introducing them.

(Page 9) “The salient concept of synaptic characteristics and the key role of memristors in modulating artificial synapses are discussed below.” for logically linking paragraph to paragraph.

(Page 11) “The following section summarizes novel and creative approaches that effectively mimic the signal processing characteristics of the above-mentioned synaptic properties in terms of their structure and architecture.” to giving concept that following two cases are for introducing novel neuromorphic design & architecture.

(4)    The authors should have supplemented with the opportunities and challenges of taking synaptic materials to applications at the end of the article, rather than simply piling up and listing the publications, a critical appraisal of the existing research should be made.

We thank the reviewer’s comments. According to the reviewer’s comments, we revise the conclusions as shown below.

Hardware-implemented neuromorphic computing based on memristor arrays was reviewed herein, including the operation mechanisms of single memristors as well as those of crossbar arrays for intelligent applications, such as pattern recognition, image processing, and AI chips. Moore’s law faces the challenge of attaining a decreasing physical scale in semiconductor processing while enhancing device performance. Emerging human-brain-inspired neuro-morphic computing aims to address the memory bottleneck associated with von Neumann architectures, which hinders memory storage and processing in big data and AI areas. Memristor-based RRAM is emerging as a promising candidate for overcoming the memory bottleneck, as it allows for high-density integration and energy-efficient memory processing using neuromorphic computing. Beyond the two-terminal memristor array, to satisfy multi-bit data storage, heterosynaptic plasticity is de-sired for tunable synaptic function, similar to that operating in the human brain. Although the memristor-based one-resistor (1R) RRAM structure is simple and capable of high-density integration, its synaptic modulation performance is poor. Therefore, one-transistor-one-resistor (1T1R) arrays should be introduced for complex AI computing and data processing tasks. Diffusive metal electrodes, such as Ag, Cu, and Ti degrade memristive switching owing to residue filament formation in the channel. Even when external-substance-induced memristive switching occurs, the degraded endurance of RRAM critically limits practical applications. Thus, in addition to cation filament formation, vacancy migration or tunneling-based synaptic devices, which generate memristive switching in-side the channel, also display significant potential for reliable neuromorphic computing. To mimic the human nerve system with memory applications, it is essential to under-stand the specific functionalities and characteristic of the nervous system, such as those of the soma, dendrites, and nodes of Ranvier. Using these components, advanced neuro-morphic computing based on realistic bio-inspired artificial neural systems ca be developed. Furthermore, integration with optic functionality broadens the potential applicability to include optogenetic and photo-induced memory. The fabrication of 2D-material-based large-scale RRAM arrays remains challenging owing to limited synthetic methods and low-yield processing. To demonstrate high-performance neuromorphic computing, fundamental advanced synthetic approaches that deliver high-uniformity in wafer-scale should be considered. The abovementioned strategies, namely, vacancy-induced migration, realistic nervous system imitation, and optogenetic integration improve the functionality and endurance of present neuromorphic computing. Memristor-based neuromorphic computing offers significant potential and functionality for in-memory processing and edge computing and should therefore be investigated further for future AI technologies.

Round 2

Reviewer 3 Report

The authors have addressed all of my concerns.